# Impact of Endotoxins in Gelatine Hydrogels on Chondrogenic Differentiation and Inflammatory Cytokine Secretion In Vitro

**DOI:** 10.3390/ijms21228571

**Published:** 2020-11-13

**Authors:** Wilhelmina M. G. A. C. Groen, Lizette Utomo, Miguel Castilho, Debby Gawlitta, Jos Malda, P. René van Weeren, Riccardo Levato, Nicoline M. Korthagen

**Affiliations:** 1Department of Clinical Sciences, Faculty of Veterinary Medicine, Utrecht University, 3584 CM Utrecht, The Netherlands; j.malda@uu.nl (J.M.); R.Levato-2@umcutrecht.nl (R.L.); N.M.Korthagen@uu.nl (N.M.K.); 2Department of Orthopaedics, Regenerative Medicine Center, University Medical Center Utrecht, P.O. Box 85500, 3508 GA Utrecht, The Netherlands; M.DiasCastilho@umcutrecht.nl; 3Department of Oral and Maxillofacial Surgery and Special Dental Care, Regenerative Medicine Center, University Medical Center Utrecht, P.O. Box 85500, 3508 GA Utrecht, The Netherlands; l.utomo-2@umcutrecht.nl (L.U.); D.Gawlitta@umcutrecht.nl (D.G.)

**Keywords:** Gelatine Methacryloyl (GelMA), mesenchymal stromal cell (MSC), articular cartilage regeneration, peripheral blood mononuclear cell (PBMC), lipopolysaccharide (LPS), type A and type B gelatine, regenerative medicine, inflammatory mediator

## Abstract

Gelatine methacryloyl (GelMA) hydrogels are widely used in studies aimed at cartilage regeneration. However, the endotoxin content of commercially available GelMAs and gelatines used in these studies is often overlooked, even though endotoxins may influence several cellular functions. Moreover, regulations for clinical use of biomaterials dictate a stringent endotoxin limit. We determined the endotoxin level of five different GelMAs and evaluated the effect on the chondrogenic differentiation of equine mesenchymal stromal cells (MSCs). Cartilage-like matrix production was evaluated by biochemical assays and immunohistochemistry. Furthermore, equine peripheral blood mononuclear cells (PBMCs) were cultured on the hydrogels for 24 h, followed by the assessment of tumour necrosis factor (TNF)-α and C–C motif chemokine ligand (CCL)2 as inflammatory markers. The GelMAs were found to have widely varying endotoxin content (two with >1000 EU/mL and three with <10 EU/mL), however, this was not a critical factor determining in vitro cartilage-like matrix production of embedded MSCs. PBMCs did produce significantly higher TNF-α and CCL2 in response to the GelMA with the highest endotoxin level compared to the other GelMAs. Although limited effects on chondrogenic differentiation were found in this study, caution with the use of commercial hydrogels is warranted in the translation from in vitro to in vivo studies because of regulatory constraints and potential inflammatory effects of the content of these hydrogels.

## 1. Introduction

Damage to articular cartilage is known to heal incompletely with natural repair resulting in the formation of fibrocartilage, which has inferior biomechanical properties compared to the original hyaline tissue. This may eventually result in gradual tissue degradation and the development of osteoarthritis (OA) [1]. For this reason, during the last decades, much effort has been put into the development of techniques to generate higher quality repair tissue. A substantial part of tissue engineering strategies use three-dimensional (3D) cultures of various cell types, including chondrocytes [2,3], mesenchymal stromal cells (MSCs) [2,4,5] and articular cartilage progenitor cells (ACPCs) [6] or combinations thereof [2,6] in a hydrogel.

Hydrogels are suitable materials to establish 3D cell cultures, as they provide a water-rich environment in which cells can thrive [7]. Gelatine, a denatured derivative of collagen, is an often-used hydrogel for the engineering of tissues. It is a natural product of animal origin from which two types are discerned, depending on tissue pre-treatment. Acid-cured tissue renders type A gelatine and with alkaline pre-treatment type B gelatine is obtained [8]. Gelatine is the base material for the manufacturing of gelatine methacryloyl (GelMA), first described by Den Bulcke et al. (2000). GelMA is a photosensitive functionalised gelatine with acrylate groups, enabling the formation of covalent crosslinks that can be triggered by, amongst others, ultraviolet (UV) or visible light exposure [9], in the presence of a photo-initiator [10]. With respect to cartilage regeneration, the material has been shown to be suitable for the 3D culturing of chondrocytes [11,12], MSCs [12,13] and ACPCs [2].

Although gelatine and GelMA are widely used in in vitro and in vivo studies, there is an important aspect that is generally overlooked: the possible influence of the presence of endogenous endotoxins in these hydrogels. This is particularly relevant as the main sources of gelatine are animal bone and skin, which are difficult to obtain through aseptic processes. Like other pathogen-associated molecular patterns (PAMPs), endotoxins may affect cell metabolism [14,15] and are able to trigger the immune system to initiate a pro-inflammatory cascade [16], which may affect several cellular functions. For example, endotoxins are suggested to play a role in the pathogenesis of OA [17,18], and in vitro cultured chondrocytes were shown to produce less collagen type II and more matrix metalloproteinase (MMP)-3 in response to lipopolysaccharides (LPS) in a dose-dependent fashion [19].

Currently, the legal maximum amount of endotoxin units (EU), imposed by all pharmacopoeias, is set to 5 EU/kg of body weight per hour for all intravenous applications of pharmaceutic and biologic products in humans [20]. For medical devices, this limit is 20 EU/device or 0.5 EU/mL when in contact with the cardiovascular and lymphatic system [21]. In the United States, the Food and Drug Administration (FDA) recommends basing the limit for medical devices on the maximum dose used in the smallest testing species [21]. For pre-clinical in vivo studies, guidelines suggest to extrapolate the human clinical amounts to the different animal species [22]. However, there has been little attention to this limit in pre-clinical animal studies as non-clinical grade materials are commonly used for in vitro studies and subsequently in preclinical animal testing without proper evaluation of the immunogenicity of these materials. Nevertheless, it is known that endotoxins, recognised by toll-like receptors (TLRs) [23], as well as pro-inflammatory factors, produced by MSCs and immune cells in response to endotoxins, can influence MSC behaviour in in vitro cultures [15,24] and in vivo research significantly. This raises potential concerns about undesired biological effects of endotoxins both in vitro and in vivo.

In this study, we tested the hypothesis that the endogenous endotoxin content of various gelatines, used to produce GelMA, would influence chondrogenic differentiation of MSCs in vitro and that it could potentially affect the innate immune system by inducing the production of pro-inflammatory mediators.

To this end, we first determined the endotoxin content of different commercially available (type A and B) gelatines and tested the feasibility of synthesising GelMA in an endotoxin-low fashion. Then we evaluated the difference in chondrogenic differentiation and cartilage-like matrix production of equine MSCs cultured in high and low-endotoxin GelMA, as well as the production of the inflammatory cytokines tumour necrosis factor-alpha (TNF-α and chemokine (C–C motif) ligand (CCL)2 for these conditions by peripheral blood mononuclear cells (PBMCs), to model the innate immune response.

## 2. Results

### 2.1. Characterisation of Gelatines and GelMAs

To determine the endotoxin levels and to check if endotoxins were introduced during functionalisation, the endotoxin content was measured in all gelatines and GelMAs (Figure 1B). No significant change in the endotoxin content was seen after GelMA synthesis and endotoxin content was high in the gelatines and GelMAs A1 and A2 and low in gelatines and GelMAs B1, A3 and B2 (in order of decreasing endotoxin level). The endotoxin content in all high-endotoxin gelatines and GelMAs was significantly different from the low-endotoxin gelatines or GelMAs (*p* < 0.0001). There was also a significant difference between GelMA B1 and B2 (*p* = 0.03). Degree of methacrylation (DM) was determined for all GelMAs: A1 67%, A2 69%, B1 74%, A3 48% and B2 84% (Figure 1A). The endotoxin level did not correlate with DM or with other gelatine or GelMA properties. There was no significant difference in compressive tangent modulus between the GelMAs, with all values within the 10 to 30 kPa range (Figure 1C). The tangent modulus of all GelMAs correlated with molecular weight (Mw) (ρ = 1.0, *p* = 0.017), but not with viscosity and Bloom strength of the different gelatines (ρ = 1.0, *p* = 0.083 and ρ = 0.5, *p* = 0.450, respectively).

### 2.2. MSC Cartilaginous Matrix Production

Histological and biochemical analyses showed that cartilage matrix formation was supported by all GelMAs (Figure 2). On day one, all hydrogels stained green in the safranin-O staining, with cell nuclei in dark blue, showing no staining for glycosaminoglycans (GAGs). On day 28, all samples showed pink areas, representing GAG formation, mainly located pericellularly (Figure 2A). Semi-quantitative assessment of the pink stained areas on day 28 revealed no significant differences between the individual GelMAs. However, when the results for the high and low-endotoxin GelMAs were grouped, significantly more pink stained surface was seen in low-endotoxin GelMAs compared to high-endotoxin GelMAs (4.52 ± 0.86% vs. 16.71 ± 3.05%, *p* = 0.013, Figure 2B). There was no significant difference in area coverage between type A and type B GelMAs (5.16 ± 4.10% and 16.07 ± 5.73% respectively; *p* = 0.177). The surface area coverage of pink Safranin-O staining correlated significantly with endotoxin level (ρ = −0.94, *p* = 0.017) and compressive tangent modulus (ρ = 0.90, *p* = 0.028).

The area staining positive for collagen type I and II was not significantly different among the GelMAs (Figure 2), while the total surface area coverage of collagen type I staining was influenced by GelMA type (A vs. B; *p* = 0.012). The surface area coverage of collagen type II correlated negatively and significantly with endotoxin level (ρ = −0.89, *p* = 0.033) and with the compressive tangent modulus (ρ = 0.84, *p* = 0.044).

No significant difference was seen in the number of GAGs between the GelMAs on day one (Appendix A, Figure A1) or on day 28 (Figure 2G). GAG content did not significantly correlate with the endotoxin level (ρ = −0.83, *p* = 0.058) or compressive tangent modulus (ρ = 0.81, *p* = 0.072; Figure 2H,I).

### 2.3. mRNA Expression

No significant differences were observed for the expression levels of genes encoding for matrix proteins (*COL1A1*, *COL2A1*), genes encoding for collagenases (*MMP3*, *MMP13*), and the gene encoding for the endotoxin receptor (*TLR4*) between the GelMAs with encapsulated MSCs at day 28. Overall, the mean levels increased between day 1 and 28 for *COL1A1* (*p* = 0.007) and *COL2A1* (*p* = 0.011), while for the other genes there was no difference over time. Post hoc analysis showed that the levels of collagen type II expression increased over the course of 28 days for GelMA A3 (34.91, 95% CI (4.80, 65.01), Appendix A, Figure A2).

### 2.4. PBMC Cytokine Production

All GelMAs were compared for TNF-α and CCL2 production by PBMCs (Figure 3). TNF-α production by PBMCs was significantly elevated for the GelMA A1 compared to all other GelMAs except for A2 (A3: *p* = 0.044; B1 *p* = 0.037; B1+: *p* = 0.045; B2 *p* = 0.036; Figure 3A). CCL2 production in response to GelMA A1 was significantly higher than the CCL2 production in response to the low-endotoxin GelMAs (A3: *p* = 0.021; B1 *p* = 0.021; B2 *p* = 012; Figure 3B). The TNF-α and CCL2 production correlated significantly with the endotoxin level (ρ = 0.94, *p* = 0.017 and ρ = 1.00, *p* = 0.003, respectively).

## 3. Discussion

This study was prompted by the fact that there is little attention for and awareness of the endotoxins endogenously present in commercially available gelatines and GelMAs, despite the widespread use of these materials in tissue engineering [7,25,26,27]. In fact, the endotoxin level is often not stated in publications, nor in the certificate of origin of the product used, resulting in the use of high-endotoxin GelMA in vivo [28,29,30]. To identify a factor that could possibly influence both in vitro cultures and the translation from in vitro to in vivo, we aimed to test the hypotheses that endogenous endotoxin content of GelMA would (a) negatively affect the chondrogenic differentiation of MSCs in vitro and (b) would induce production of pro-inflammatory mediators that could potentially affect the innate immune system.

Endogenous endotoxins present in collagen hydrogels are extremely difficult to remove, possibly due to the complexes they form with procollagen [31]. This is illustrated by the fact that in two of the commercially available and widely used gelatines included in this study, a high endotoxin level was found, which was not altered by dialysis. The endotoxin level of these hydrogels far exceeds the legal threshold for medical devices [21], indicating that high endotoxin levels can indeed be a concern. Low endotoxin levels were found in three of the gelatines included in this study, showing that low-endotoxin alternatives are commercially available for use in both in vitro and in vivo studies.

### 3.1. Effect of Endotoxins

Endotoxins have been suggested to play a role in the pathogenesis of OA through activation of macrophages [17]. We have shown that PBMCs, when exposed to GelMAs with a high endogenous endotoxin content, can produce significantly higher amounts of TNF-α and CCL2. TNF-α and CCL2 (also referred to as monocyte chemoattractant protein (MCP)-1) are two of the many cytokines that can be produced in the body under pro-inflammatory conditions [32,33,34], such as in the presence of endotoxins or other PAMPs. Both cytokines have been shown to induce detrimental effects on many tissues in the joint [35,36,37,38] and this could theoretically compromise the functionality of an implant [39].

It has been shown that TNF-α can suppress chondrogenesis [40] and plays an important role in matrix degradation, as it stimulates resorption and inhibits synthesis of proteoglycan in cartilage [41]. CCL2 has been reported to promote collagen and aggrecan proteolysis in chondrocyte cultures and cartilage explants [42]. It also recruits monocytes that seem to propagate inflammation and tissue damage in osteoarthritis [43] and rheumatoid arthritis [44]. Thus, the fact that high-endotoxin GelMAs can induce TNF-α and CCL2 production is a strong motivation to question their use for in vivo cartilage regeneration.

Although cytokine production of PMBCs was influenced by the high level of endotoxins in GelMA A1, in vitro chondrogenic differentiation of equine MSCs was similar in high- and low-endotoxin GelMAs. The MSCs, embedded in GelMA discs and cultured in chondrogenic differentiation medium, produced comparable cartilage-like matrix in terms of GAG, collagen type II and collagen type I, in both the high- and low-endotoxin GelMAs used in this study. Further, all GelMAs supported the production of collagen type II and collagen type I by the embedded MSCs over time and there was no significant difference between the groups. On the other hand, there seemed to be a correlation between endotoxin level and histological staining for glycosaminoglycans and collagen type II. Nevertheless, these results suggest that, with the set-up used in this study, endogenous endotoxin level is not a very critical factor when it comes to the chondrogenic matrix production of the embedded MSCs in vitro.

For this study, we used equine bone marrow-derived (BM-)MSCs, because the horse is both a patient with a large clinical need for better articular cartilage repair and a recommended animal model for pre-clinical testing of cartilage regeneration strategies [45,46]. Further, equine MSCs do express the receptor that allows endotoxin signalling, namely TLR4 [47,48], and have previously been shown to be sensitive to LPS [49]. Our data suggest that the presence of endogenous LPS does, however, not affect chondrogenic differentiation of healthy equine BM-MSCs embedded within GelMA hydrogels. Nevertheless, for human MSCs, contrasting reports can be found on the effects of TLR ligation [50,51]. For example, chondrogenic differentiation of human MSCs was found to be either unaffected [50,51] or inhibited by a 1000 times higher dose of LPS [52]. The reported effects on MSC proliferation are also contradictory; LPS was found to stimulate MSC proliferation in a concentration-dependent fashion [23] with the stimulatory effect already present at low-endotoxin concentrations [35], while others could not find any effect of LPS on MSC proliferation [35,36,37,38]. These contrasting reports on MSC response to LPS could be due to differences in species, tissue source and donor age [15,50,53]. Another possibility is that the chondrogenic medium that we used contained low concentrations of dexamethasone that may have mitigated to a certain extent the negative effect of the endotoxins [54].

### 3.2. Effect of Gelatine Characteristics

The influence of gelatine characteristics, such as the GelMA type and stiffness, were not included in the aims of this study, as obtaining multiple commercially available gelatines from different companies with varying endogenous endotoxin content, but with identical physical characteristics is not feasible. Nevertheless, a difference was observed between type A and B GelMAs in the production of collagen type I. Gelatine types A and B are produced in different ways [55], which results in different isoelectric points [56] and can influence the degree of methacrylation (DM) [43] and molecular weight [55]. Furthermore, the species and tissue type from which the gelatine is extracted could influence cell response [47,48]. Type A and B variants from bovine and porcine origin were included, but as the current study was not designed to identify the influence of gelatine type, further research is needed to elucidate which qualities of type A and type B GelMA could influence MSC differentiation and thereby impact tissue engineering and regenerative medicine applications.

The stiffness of scaffolds is also known to influence a wide variety of cell responses, including the differentiation capacity of MSCs [57,58]. We aimed to synthesise GelMAs with comparable stiffnesses in the appropriate range for chondrogenic differentiation The DM of GelMA A3 was, however, lower than the other GelMAs. Despite the lower DM, there were no statistically significant differences in compressive stiffness between our hydrogels. The lower DM of GelMA A3 was most likely compensated by the higher bloom strength of this gelatine. The tangent moduli ranged from 10 to 30 kPa, which is within the range deemed suitable for chondrogenic differentiation in earlier studies [57,59].

### 3.3. Synthesis of Low-Endotoxin GelMA

Low-endotoxin gelatines can be obtained by treating animal tissue and/or gelatine with different substances, e.g., alcohol and/or acetone at a pH of 10 to 12, calcium hydroxide and a quaternary ammonium salt at a pH of 12 or Triton X-100 and an absorbent like active coal [60]. Furthermore, ultrafiltration through a membrane of maximum 100 kDa can render a reduction of endotoxins [61], however, all of these methods are patented [60] and do not remove endotoxins which are bound to the gelatine macromer itself.

We have shown in this study a straightforward method of synthesising GelMA that does not introduce additional endotoxins compared to the basic gelatine endotoxin levels. This means that, if the base material (gelatine) has a low endotoxin content and the cleaning and handling of the product are carefully executed, the endotoxin level of the end product will be acceptable. In fact, the endotoxin content of three out of five gelatines used in this study was lower than 10 EU/g, which means that these gelatines could comply with the legal threshold for a medical device which is 20 EU/device [21].

### 3.4. Limitations

There are various methods available to quantify endotoxin levels [16,62]. The method chosen for testing the endotoxin content in this study, the LAL chromogenic assay, is sensitive for low amounts of endotoxins with a detection range from 0.1 to 1 EU/mg, and is associated with a wide range in the coefficient of variation [16]. To account for this, repeated measurements were performed. Another possible limitation is that it is difficult to make a valid comparison between our study and other in vitro studies using endotoxins such as LPS. Dose and timing of LPS stimulation (e.g., continuous or priming/pre-conditioning) can cause alterations in MSC response. In addition, EU values can differ between different LPS sources and batches, and EU values are generally not mentioned in the literature. Furthermore, continual exposure to endotoxins from the surrounding hydrogel is very different from adding LPS to the culture medium and refreshing twice a week. Lastly, GelMA itself has been shown to be able to sequester TNF-α released by monocytes and this phenomenon may have resulted in an underestimation of the cytokines produced by PBMCs in this study [63].

## 4. Materials and Methods

### 4.1. GelMA Synthesis

Five different gelatines (A1, A2, A3, B1 and B2; Figure 1A) were used to synthesise GelMA according to the method described by Melchels et al. [64]. An excess of methacrylic anhydride (MA, ≥92%; Sigma-Aldrich, St. Louis, MO, USA) was added to gelatine (0.6:1) in sterile phosphate-buffered saline (PBS) at 50 °C for 1 h. Excess MA was removed by centrifugation and decanting. The pH was adjusted to 7.4 using NaOH and/or HCL. This was followed by dialysis (cellulose membrane 14 kDa cut off; Sigma-Aldrich) against MQ-water at 4 °C for five to seven days. MQ-water was refreshed daily. Afterwards, GelMA solutions were sterile-filtered and subsequently freeze-dried and stored at −20 °C. Prior to use, all non-sterile tools were cleaned with acetone and 70% ethanol and then heated to 250 °C for 30 min or 180 °C for 2.5 h to remove endotoxins [20].

### 4.2. Determination of Degree of Methacrylation

The DM was calculated according to Hoch et al. [65]. Lyophilised gelatines and GelMAs were dissolved in D_2_O at a concentration of 10 mg/mL. The DM was defined as the percentage of ε-amino groups of gelatine (lysine, hydroxylysine) that are substituted with acryloyl moieties. The protons of these moieties give signals in the lysine methylene region at 2.7 to 2.8 ppm. For the quantification of the DM by ^1^H NMR the spectra were normalised to the phenylalanine signal in the region 7.2 to 7.3 ppm, which represents the backbone of the gelatine. The normalised lysine methylene signals (2.7 to 2.8 ppm) of gelatine and GelMA represent the areas *A*(lysine methylene of unmodified gelatine) and *A*(lysine methylene of GelMA). The DM of the different GelMAs was calculated as:$$\mathrm{DM}[\%]=(1-\frac{A(\mathrm{lysine}\;\mathrm{methylene}\;\mathrm{of}\;\mathrm{GelMA})}{A(\mathrm{lysine}\;\mathrm{methylene}\;\mathrm{of}\;\mathrm{unmodified}\;\mathrm{gelatine})})\times 100\%$$

### 4.3. Endotoxin Levels of the Gelatines and Synthesised GelMAs

A concentration range (depending on expected endotoxin level) was prepared for solutions of all five gelatines and GelMAs. The Pierce^TM^ LAL Chromogenic Endotoxin Quantitation Kit (Thermo Fisher Scientific, Waltham, MA, USA) was used to determine endotoxin content, following the manufacturer’s instructions. The chromogenic test is a method approved by the European Pharmacopoeia. The endotoxin standard provided in the kit was matched to comply with FDA requirements. Measured values coincided with the gelatine manufacturer’s measurements; therefore, interference testing was not performed. Endotoxin level was considered low when a value of <10 EU/g was measured. If this value was higher than 100 EU/g the level was considered high. Values in between these limits were considered intermediate. Measurement of gelatine B2 was repeatedly below detection level and was set to the lowest detectable value (0.2 EU/mL for a 500 mg/mL solution).

### 4.4. Hydrogel Preparation—Cell Free

Irgacure 2959 (BASF, Ludwigshafen, Germany) was dissolved in PBS-MQ at 0.1% *w*/*v* at 70 °C for 5 to 10 min. All GelMAs were dissolved separately in PBS-Irgacure at 10% *w*/*v* on a roller bench at 37 °C for 30 min. GelMAs were cast in discs (6 mm Ø × 2 mm) using custom made Teflon moulds at room temperature (RT) and UV crosslinked for 15 min (UVP CL-1000 Ultraviolet Crosslinker, intensity 6.4 mW cm^−2^).

### 4.5. Mechanical Testing

The tangent moduli of cell-free hydrogels were determined with uniaxial unconfined compression using dynamical mechanical analysis (Q800; TA Instruments, New Castle, Delaware, USA). Samples were soaked in PBS for 24 h, pre-loaded at 0.001 N, followed by ramp force at a rate of 20% strain/min until 30% strain. The compressive tangent modulus was measured as the slope of the stress–strain curves in the 0 to 2.5% strain range. For all five GelMAs, three different sets of 2 to 10 technical replicates were measured, and the average value for each set was reported.

### 4.6. Isolation and Expansion of Equine Mesenchymal Stromal Cells

MSCs were isolated from bone marrow aspirates of the sternum of three healthy female Shetland ponies (donor A: 6 yrs; donor B: 4 yrs; donor C: 11 yrs), with approval of the local animal ethical committee. Bone marrow was diluted in PBS, filtered through a 100 µm cell strainer, and pipetted onto a layer of Ficoll-Paque PLUS. The cells were centrifuged, and the mononuclear cell layer was transferred to a fresh tube. The cells were washed and cultured in monolayer (0.25 × 10^6^ /cm^2^) with high glucose DMEM (D6429; Sigma-Aldrich) supplemented with 10% foetal bovine serum (FBS, Gibco), 1% penicillin-streptomycin (P/S, final concentration of 100 units/mL penicillin and 100 µg/mL streptomycin, Gibco), and 1 ng/mL recombinant human fibroblast growth factor-basic (bFGF, Peprotech, Rocky Hill, NJ, USA). The selection of MSCs was based on plastic adherence and the cells were stored in liquid nitrogen at passage three until further use. Upon thawing, MSCs were cultured in monolayer with a seeding density of 0.2 × 10^4^ /cm^2^ to reach passage four for further use.

### 4.7. MSC Encapsulation and Culture

Irgacure was dissolved in PBS-MQ at 0.125% *w*/*v* at 70 °C for 5 to 10 min. All five GelMAs were dissolved separately in PBS-Irgacure at 12.5% *w*/*v* on a roller at 37 °C for 30 min. MSCs were trypsinised, suspended in PBS-MQ and encapsulated in each GelMA separately yielding a final concentration of 10% *w*/*v* and 20 × 10^6^ cells/mL. No high-endotoxin alternative was available for B1, hence, to fabricate a high-endotoxin B1+, LPS from *Escherichia coli* 055:B5 ((L5418); Sigma-Aldrich) was added to the hydrogel at a concentration of 36.6 EU/mL, resulting in a sixth hydrogel solution. These six hydrogel solutions were cast in discs (d = 6 mm; h = 2 mm) using custom-made Teflon moulds and UV crosslinked for 15 min. The Teflon moulds were cleaned as described above. Twenty discs per donor were cultured for each hydrogel, 10 were used for histological and biochemical analysis and 10 for gene expression analysis. Out of 10 discs, four were used for day 1 and six for day 28. The discs were cultured in chondrogenic differentiation medium consisting of DMEM (41965; Gibco) supplemented with 1% P/S, 0.2 × 10^−6^ /mL l-ascorbic acid-2-phosphate (Sigma-Aldrich), 1% insulin-transferrin-selenous acid (ITS+ Premix; Corning), 0.1 × 10^−6^ /mL dexamethasone (Sigma-Aldrich), and 10 ng/mL recombinant human transforming growth factor ß1 (Peprotech, UK) for 1 and 28 days. The medium was refreshed twice per week.

### 4.8. Biochemical Analysis of Cultured GelMA-MSC Constructs

After chondrogenic differentiation, the GAG and DNA content of the hydrogels was determined to evaluate cartilage-like matrix formation. After 1 and 28 days of culture, the GelMA discs were cut in half. One half of each disc was weighed, freeze-dried and weighed again prior to digestion using 200 µL papain digestion buffer (0.2 M NaH_2_PO_4_ + 0.01 M EDTA, pH = 6.0, supplemented with 250 µg/mL Papain (P3125, Sigma-Aldrich) and 1.57 mg/mL Cysteine HCL (C9768, Sigma-Aldrich) overnight at 60 °C.

For determination of GAG concentration, papain-digested samples were diluted in PBS-EDTA and 46 µM 1,9-Dimethyl-Methylene Blue (DMMB, Sigma-Aldrich) solution was added to the samples. Absorption was measured at 525 and 595 nm on a spectrophotometer [66] and the ratio was used to calculate the concentration with a chondroitin sulphate C (Sigma-Aldrich) standard curve.

DNA concentration was measured in the papain digests with Picogreen (Quant-iT, Thermo Fisher Scientific) according to the manufacturers’ instructions. In short, samples were diluted 1:20 in TE-buffer (0.5 M EDTA in 1 M Tris (1:5), pH 8) and 100 µL sample was mixed with 100 µL Picogreen reagent, incubated 5 min in the dark and fluorescence was measured with an excitation of 485 nm and emission of 520 nm. Known concentrations of λDNA were used as a reference.

### 4.9. Histology and Immunohistochemistry

The other half of all GelMA discs cultured with encapsulated MSCs were fixed in 4% formaldehyde, dehydrated using a graded ethanol series and cleared in xylene. Next, samples were embedded in paraffin and sections were cut with a thickness of 5 µm.

Sections were stained with safranin-O, fast green and haematoxylin to stain proteoglycans pink, collagens green and nuclei purple. To visualise collagen type I and II using immunohistochemistry, sections were deparaffinised in xylene, hydrated with graded ethanol steps and blocked in 0.3% H_2_O_2_/PBS. Next, antigens were retrieved using 1 mg/mL pronase/PBS (11459643001; Roche, Basel, Switzerland) and 10 mg/mL hyaluronidase/PBS (H2126; Sigma-Aldrich) both for 30 min at 37 °C. Sections were blocked with PBS-BSA 5% *w*/*v* for 30 min at RT and incubated with the primary antibodies (collagen type II, mouse monoclonal antibody, DSHB, II-II6B3, 1:100; collagen type I, rabbit monoclonal antibody, Abcam, ab138492) overnight at 4 °C. Mouse monoclonal IgG1 antibody (DAKO, X0931, 1:100) was used as a negative control. Subsequently, the sections were incubated with the secondary antibody (goat anti-mouse IgG HRP, DAKO P0447, 1:100, EnVision+; System-HRP anti-rabbit DAKO, K4003) for 60 min at RT and stained using DAB peroxidase substrate solution (Sigma-Aldrich) for 5 to 10 min. Counterstaining was performed using Mayer’s haematoxylin and sections were dehydrated with graded ethanol steps, cleared in xylene and mounted with Depex (100579, Merck Millipore, Billerica, MA, USA). For each staining, all sections were stained at the same time and randomised over different jars. The sections were evaluated and photographed using a light microscope (Olympus BX51, Hamburg, Germany). The relative surface of the pink area in the safranin-O-stained sections and of the brown area in collagen II and collagen I-stained sections were analysed semi-quantitatively using Image J (version 2.0.0-rc-691.52p). Images of a representative area were converted to 8-bit. A threshold was determined manually, and pixels were converted to black and white. The same threshold was used for all sections of the same staining. Of each section, the number of black pixels was determined by Image J and manually divided by the total number of pixels of the tissue section.

### 4.10. Equine PBMC Isolation and Culture

Blood samples (30 mL) of three healthy, skeletally mature female Shetland ponies (donor 1: 3 yrs; donor 2: 15 yrs; donor 3: 3 yrs) were diluted 2.5× in PBS-MQ, and 35 mL of this suspension was carefully added to 15 mL Ficol-Paque PLUS (1.077 g cm^−3^, GE Healthcare) and centrifuged at RT, 400× *g* for 30 min without brake. The mononuclear cell layer was carefully aspirated and transferred to a new 50 mL conical tube and washed twice with DMEM 41965 (Gibco). Cells were counted and frozen in freezing medium (DMEM containing 10% dimethyl sulfoxide (DMSO; Merck, Burlington, MA, USA), and 20% FBS) and stored in liquid nitrogen until further use.

Cell-free GelMA discs were fabricated as described in the scaffold fabrication section, again including GelMA B1+ to provide a high-endotoxin alternative for B1. For cell culture purposes, Teflon moulds were cleaned as described above. PBMCs were seeded on top of cell-free GelMA discs (*n* = 5 per group) with a density of 2.5 × 10^5^ /mL and cultured in 0.5 mL RPMI-1640 + GlutaMAX (Gibco) supplemented with 10% FBS and 1% P/S for 24 h in 48-wells plates. After culture, the medium was removed, centrifuged for 5 min at 500× *g* and the supernatants were stored separately in Eppendorf tubes at −80 °C until protein quantification. The hydrogels and attaching cells were stored in 350 µL RLT lysis buffer (Qiagen, Hilden, Germany) supplemented with 10 μL/mL β-mercaptoethanol (Sigma-Aldrich, St. Louis, MO, USA) at −80 °C until further processing for DNA quantification.

### 4.11. Biochemical Analysis of PBMC Culture

Protein concentrations of TNF-α (TNF alpha Equine Uncoated ELISA Kit, ESS0017; Thermo Fisher) and CCL2 (Equine CCL2 (MCP-1) Do-It-Yourself ELISA, DIY0694E-003; Kingfisher Biotech) of the PBMC culture supernatants were measured by enzyme-linked immunosorbent assays (ELISAs). Briefly, high binding 96-well plates (EIA/RIA; Costar #9018) were coated with 100 µL coating antibody (1 µg/mL in carbonate-bicarbonate buffer) and incubated overnight at RT. The next morning, the plates were blocked with 300 µL PBS-BSA 1% for 90 min and then 100 µL of standard or supernatants diluted in RPMI culture medium (1:1 for TNF-α, 1:20 for CCL2) were incubated for 90 min at RT. Plates were washed four times using PBS with 0.1% Tween 20 (Sigma), incubated with 100 µL detection antibody (TNF-α 1:100 and CCL2 0.1 µg/mL in PBS-BSA 1%) for 60 min at RT, washed four times, and incubated with 100 µL Streptavidin-HRP (20 ng/mL in PBS-BSA 1%, 21134; Thermo Scientific) for 30 min at RT. Finally, after washing four times, plates were incubated with 100 µL TMB Substrate solution (N600; Thermo Scientific) for 30 min in the dark and then the reaction was stopped by adding 100 µL 0.18 M H_2_SO_4_. Absorbance was measured at 450 nm and a background measurement at 550 nm was subtracted. A standard curve was plotted using a four-parametric curve. Samples that were below the lowest value in the standard curve were set to the lowest value of the standard curve.

To normalise the protein concentrations, DNA concentrations of the PMBCs were measured with a Picogreen assay (Thermo Fisher) on the cells in RLT buffer [67]. Optimal dilution of 1:40 was determined. GelMA discs present in the RLT-buffer were crushed using pistons cleaned in RNaseZAP (Sigma), RNAse-free 70% ethanol and RNAse-free H_2_O. Samples were then vortexed, centrifuged at 500× *g* for 1 min and 10 µL was used to dilute 1:40 in TE-buffer. Picogreen was performed as described above, except standards that were dissolved in TE-diluted RLT buffer (1:40) to match samples.

### 4.12. qRT-PCR

To evaluate if the alteration in mRNA expression occurred due to the different GelMAs, PCR analysis was performed. Separate discs with encapsulated MSCs were cultured for 1 and 28 days to analyse gene expression. Samples stored at −80 °C in RLT lysis buffer supplemented with β-mercaptoethanol were thawed and crushed using pistons. Subsequently, mRNA was isolated using the RNeasy Mini kit (Qiagen, Germany) according to the manufacturer’s instructions. Amplification and cDNA synthesis were performed using the SuperScript III Platinum SYBR Green One-Step qRT-PCR Kit (Invitrogen, Thermo Fisher). The relative expressions of *COL1A1*, *COL2A1*, *MMP3*, *MMP13* and toll-like receptor (*TLR)4* were analysed and normalised to the expression of housekeeping gene *HPRT1* (for primer sequences see Appendix A, Table A1). Relative expressions were calculated from efficiency and Ct values extracted using the PCR Miner algorithm [68].

### 4.13. Statistical Analysis

SPSS (version 25, IBM Corporation, Armonk, New York, NY, USA) was used for statistical analysis. Normality was evaluated using Shapiro–Wilk and data were log-transformed to normalise the distribution. The change in endotoxin level before and after synthesis was compared using a repeated-measures ANOVA and the endotoxin content of gelatines and GelMAs were compared using a one-way ANOVA. The compressive tangent moduli of different GelMAs were compared using the Kruskal–Wallis test. GAG and cytokine production normalised to DNA was compared using a linear mixed model, using GelMA as a fixed factor and donor as a random factor to take into account donor variation. Results of histology and PCR data were analysed using a two-way ANOVA or Friedman test. For a grouped comparison of high and low-endotoxin groups and their respective type A and type B GelMA, a paired t-test was used. To correct for multiple comparisons in the ANOVAs and linear mixed models, the Sidak post-hoc test was used for parametric tests and Dunn’s post-hoc test was used for the non-parametric analysis. All correlations were calculated using Spearman’s correlation coefficient. Differences of *p* < 0.05 were considered significant.

## 5. Conclusions

This study showed that the varying endotoxin levels, as present in the hydrogels in this study, were not a critical factor for the in vitro chondrogenic differentiation of MSCs encapsulated in GelMA. However, endogenous endotoxins present in gelatines were shown to induce an in vitro inflammatory response in PBMCs that might produce negative effects when constructs are placed in a more complex, multi-cellular environment, as in co-culture experiments or even in vivo. Therefore, careful selection of low-endotoxin variants of gelatine and GelMA is recommended in order to minimise potential pro-inflammatory responses upon in vivo implantation, which may detrimentally affect cartilage repair. In addition, choosing gelatines and GelMAs with endotoxin contents that comply with the legal limits for clinical use of biomaterials will improve consistency and logic in the translation from bench to bedside.

## Figures and Tables

**Figure 1 ijms-21-08571-f001:**
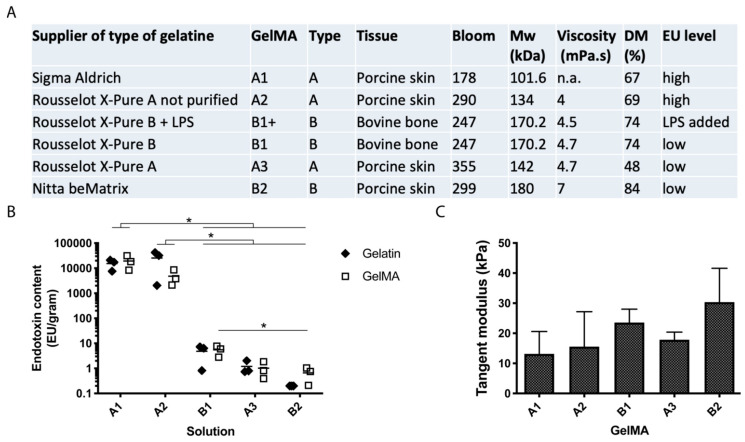
Gelatine and GelMA characteristics. (**A**) Data of gelatines and their GelMA derivatives used in this study, including specifications supplied by the fabricating company (type, tissue, bloom strength, Mw, viscosity) and additional measurements performed in this study (DM, EU level). (**B**) Endotoxin content of different gelatine and GelMA solutions. GelMAs are shown in order from high to low endotoxin content. (**C**) Compressive tangent moduli of hydrogels after crosslinking. Values represent mean + SD of independent experiments (*n* = 3) with GelMA discs (2 to 10 technical replicates). *: *p* < 0.05.

**Figure 2 ijms-21-08571-f002:**
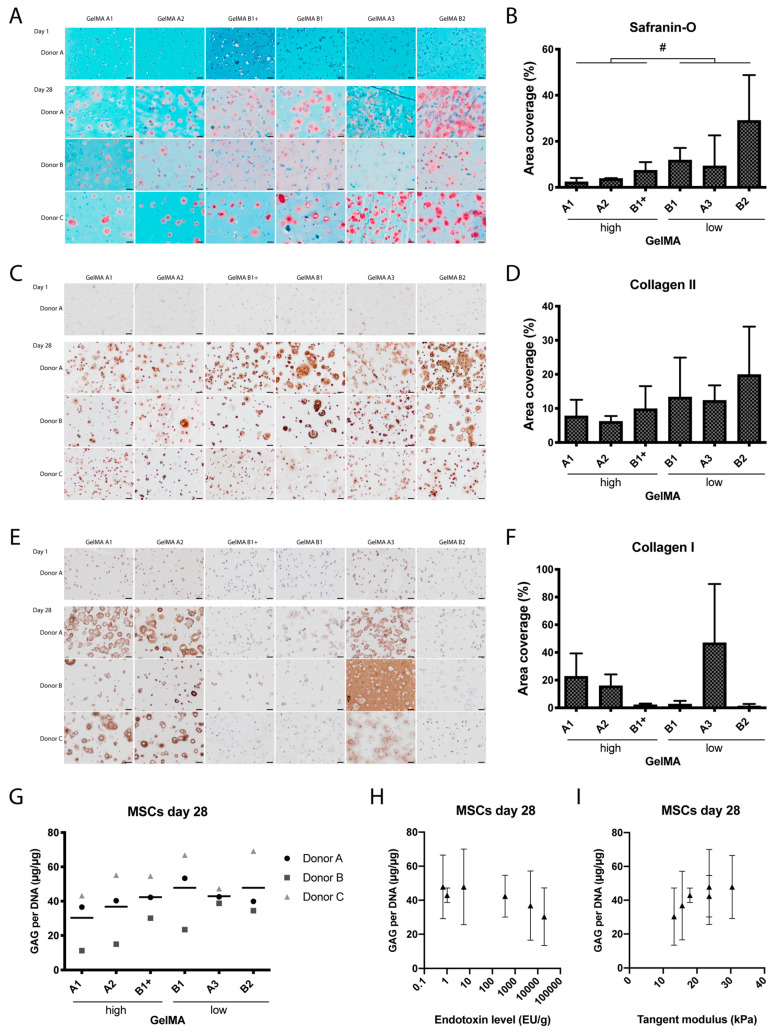
Cartilage-like matrix formation by equine MSCs after 28 days of culture in six hydrogels from five different GelMAs. GelMA B1 was supplemented with LPS to obtain high endotoxin GelMA B1+. Baseline immunohistochemical stainings are shown for donor A on day one and for all donors on day 28. Safranin-O/Fast Green stains collagen, bone or cytoplasm green and GAGs red/pink (**A**). Immunohistochemical staining for collagen type II (**C**) and collagen type I (**E**) in brown; nuclei are stained blue. Scale bars represent 50 μm. Semi-quantitative analysis of the total surface of pink stained areas in the safranin-O staining (**B**) and of brown stained areas in the collagen type II (**D**) and collagen type I (**F**) immunostainings. GAG content normalised to DNA after 28 days of chondrogenic differentiation (**G**). GAG content did not correlate significantly with the endotoxin level (**H**) or the compressive tangent modulus (**I**). The symbols in G represent the mean of six technical replicates per donor, the line represents the mean of three donors. In (**B**,**D**,**F**,**H**,**I**) the values represent mean ± SD of three donors. #: significant difference of grouped analysis (*p* < 0.05).

**Figure 3 ijms-21-08571-f003:**
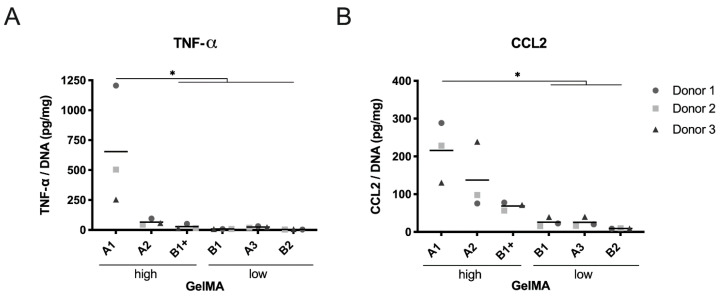
Cytokine production of PBMCs in response to GelMA hydrogels. TNF-α (**A**) and CCL-2 (**B**) protein levels in the culture supernatant of PBMCs after a 24-h culture on six hydrogels from five different GelMAs. GelMA B1 was supplemented with LPS to obtain high endotoxin GelMA B1+. *n* = 3 PBMC donors. Symbols represent the mean of five technical replicates per donor and are expressed in pg protein/mg DNA. *: *p* < 0.05.

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
