# Peer review of "Impact of Endotoxins in Gelatine Hydrogels on Chondrogenic Differentiation and Inflammatory Cytokine Secretion In Vitro"

_ijms, 2020, doi:10.3390/ijms21228571_

Round 1

Reviewer 1 Report

This work evaluates the endotoxin level and chondrogenic differentiation of five different GelMA hydrogels. The authors reported varied contents of endotoxin between types of GelMA but no significant differences were observed in matrix production of equine MSCs. GelMA with higher endotoxin levels produced more TNF-α and CCL2 than that of lower endotoxin hydrogel. Hydrogels are widely used in regenerative medicine, however, the immunological reaction of hydrogels have not yet been fully characterized. This study provides some evidence that the inflammatory responses from GelMAs may be correlated with endotoxin levels. The study is well designed and written. Please elaborate on the rationale for mechanical testing. Please also discuss possible solutions to reduce the endotoxin levels and inflammatory responses.

Author Response

  • Please elaborate on the rationale for mechanical testing.

         Mechanical testing is performed to ensure that the stiffness of the hydrogels is in the appropriate range for cartilage matrix production and to ensure that there are no large differences between the different hydrogels evaluated. It has been established that MSCs are mechanosensitive cells and stiffness of surrounding hydrogel matrix can drive these cells towards specific fates.

         For example, a higher stiffness could lead towards differentiation of the embedded MSCs into the osteogenic lineage, whereas a lower stiffness could result in differentiation into the adipogenic, neurogenic or myogenic lineages. (Mousavi 2015; Huebsch 2010; Wang 2010; Chaudhuri 2015)

Adapted line 243-246:

The stiffness of scaffolds is also known to influence a wide variety of cell responses, including the differentiation capacity of MSCs[57,58]. We aimed to synthesise GelMAs with comparable stiffnesses in the appropriate range for chondrogenic differentiation to avoid matrix-driven differentiation into other lineages.

  •  Please also discuss possible solutions to reduce the endotoxin levels and inflammatory responses.

LPS molecules (endotoxins) are about 10 kDa in size, but can form large aggregates in aqueous media, also named "micelles" having a molecular weight of up to 1000 kDa. Endotoxins can be reduced or removed in multiple ways. Different companies have filed a patent for their method. Reduction of endotoxins in gelatine can be performed by ultrafiltration (Kanayama 2007, Jellice patent EP1829946). The size of the membrane limits the option of including larger molecules to produce stiffer hydrogels.Endotoxins in insoluble collagen can be reduced to less than 10 EU/g, by treating the collagen with aqueous alkali, acid and oxidizing agent without dissolving the collagen (Patent WO 2012/031916). Furthermore, collagen protein can be subjected to a basic alcohol and/or acetone treatment at a pH of 10 - 12, which decomposes the endotoxin contained in the collagen protein. The resultant protein, having less than 1000 EU/g LPS, is recovered by precipitation (Nitta, patent JP2004300077). Alternatively, animal tissue can be treated with calcium hydroxide and a quaternary ammonium salt at a pH of 12 for at least 5 days. Subsequently, the gelatine solution is neutralized to a pH of 4.5-5 with acid, and gelatine is obtained by extraction at a temperature of > 65 °C. The gelatine can further be sterilized by filtration through a 0.2 micrometer membrane containing less than 5 EU/g of endotoxins (Nippi, patent JP2005289841). This method limits the molecular weight up to 200KDa because of the pore size of the membrane used in the filtration step. Furthermore, the high pH used in the last two methods results in deamination of the gelatine, causing a decrease in isoelectric point to 5 - 6, i.e. only gelatine of type B can be obtained. Lastly, endotoxins can be removed from an aqueous gelatin preparation using a micelle forming surfactant such as Triton X-100. Next, adsorption of the surfactant and LPS can be performed by a solid adsorbent, such as active coal. This method does not render temperature elevation, enabling removal of LPS under gentle conditions, and keeps the properties of the gelatine such as viscosity virtually similar to those of the gelatine before the LPS removal (Rousselot patent WO2016085345A1).

Added line 252-257:

Low-endotoxin gelatines can be obtained by treating animal tissue and/or gelatine with different substances, e.g. alcohol and/or acetone at a pH of 10-12, calcium hydroxide and a quarternary ammonium salt at a pH of 12 or Triton X-100 and an absorbent like active coal (Patent Rousselot, 2016). Furthermore, ultrafiltration through a membrane of maximum 100 kDa can render a reduction of endotoxins (Kanayama, 2007), however all of these methods are patented (Patent Rousselot, 2016) and do not remove endotoxins which are bound to the gelatine macromer itself.

Reviewer 2 Report

The authors investigated the endotoxin level of five different GelMAs and evaluated its impact on the chondrogenic differentiation of equine mesenchymal stromal cells (MSCs). Overall, the study will provide the required knowledge in the GelMAs based biomaterials selection for diverse biomedical applications, and the manuscript is suitable for publication with minor modification.

Major comments.

The bacteria and the contamination of its products with biomaterials would cause a serious issue if the contaminated materials exposed to the systemic circulation.  In general, bacterial endotoxins test is used to detect or quantify bacterial endotoxins of gram-negative bacterial origin using an amoebocyte lysate prepared from blood corpuscle extracts of horseshoe crab (Limulus polyphemus or Tachypleus tridentatus). The authors used a commercial kit (The Pierce 294 TM LAL Chromogenic Endotoxin Quantitation Kit (Thermo Fisher 295 Scientific) to determine endotoxin content that is acceptable for endotoxin testing at lab scale level. "It would be good if the authors ensure that the tested procedures and reported values are according to the Ph. Eur. Standards".

 Minor comments

The figure and table should be improved before publication.

Author Response

  • In general, bacterial endotoxins test is used to detect or quantify bacterial endotoxins of gram-negative bacterial origin using an amoebocyte lysate prepared from blood corpuscle extracts of horseshoe crab (Limulus polyphemus or Tachypleus tridentatus). The authors used a commercial kit (The Pierce 294 TM LAL Chromogenic Endotoxin Quantitation Kit (Thermo Fisher 295 Scientific) to determine endotoxin content that is acceptable for endotoxin testing at lab scale level. "It would be good if the authors ensure that the tested procedures and reported values are according to the Ph. Eur. Standards".

 The chromogenic test used in this paper to analyse bacterial endotoxin content is a method approved by the European Pharmacopoeia (10.4). All materials used for the tests are either pyrogen free culture plastics or were cleaned by a validated method (European Pharmacopoeia  10.4; Magalhães 2007). The endotoxin standard provided in the kit was “matched” to comply with Food and Drug Administration (FDA) requirements for endotoxin testing. Each LAL lot is tested for functionality using the United States Reference Standard EC-6. The LAL lot is then matched to a lot of our Control Standard Endotoxin (CSE) by testing in parallel with the Reference Standard Endotoxin (RSE). The RSE/CSE correlation assay determines the potency of each CSE lot when used with each matching LAL lot (from product information in the user guide of the LAL Chromogenic Endotoxin Quantitation Kit). 

As endotoxin levels measured in the LAL test were comparable with the values measured by the manufacturers of the low-endotoxin gelatines, no testing for interference was deemed necessary. Furthermore, Rousselot measured the GelMAs produced by the authors using the Endozyme method, which gave comparable results for the low-endotoxin GelMAs. The Endozyme method is based on recombinant factor-C and is not sensitive to b-glucan interference. This justifies the non-use of an endotoxin specific buffer. Because the high dilution, required to measure high-endotoxin gelatines and GelMAs in the narrow range of the LAL standard curve, will create higher uncertainty, no additional spiked samples were included for interference testing.

 Although PBMC culture (a version of the MAT) is a method accepted by the European Pharmacopoeia, the substances used were not tested as a solution, but in crosslinked gel form, and therefore our version of the MAT does not comply with the Ph. Eur. Standards. It was not our intention to measure endotoxin activity in a quantitative/semi-quantitative way using the MAT. We did aim to show clinical implications of endotoxin containing GelMAs.

 Adapted line 300-304:

A concentration range (depending on expected endotoxin level) was prepared for solutions of all five gelatines and GelMAs. The PierceTM LAL Chromogenic Endotoxin Quantitation Kit (Thermo Fisher Scientific) was used to determine endotoxin content, following manufacturer’s instructions. The chromogenic test is a method approved by the European Pharmacopoeia. The endotoxin standard provided in the kit was matched to comply with FDA requirements. Measured values coincided with gelatine manufacturer’s measurements, therefore, interference testing was not performed.

  •  The figure and table should be improved before publication.

High resolution figures (600dpi) in are included in the submission system, but compressed versions are included in this document for the purpose of the review process.